# Insufficient iodine nutrition status and the risk of pre-eclampsia: a systemic review and meta-analysis

Charles Bitamazire Businge [1,2] Anthony Usenbo,[3] Benjamin Longo-Mbenza,[4] AP Kengne[2,5]

¹Department of Obstetrics and Gynaecology, Faculty of Health Sciences, Walter Sisulu University, Mthatha, South Africa
²Department of Medicine, Faculty of Health Sciences, University of Cape Town, Cape Town, South Africa
³Department of Anaesthesiolgy, Nelson Mandela Academic Hospital, Mthatha, South Africa
⁴Faculty of Medicine, University of Kinshasa and LOMO University of Research, Kinshasa, Democratic Republic of Congo
⁵Non-Communicable Disease Research Unit, South African Medical Research Council, Cape Town, South Africa

**Correspondence to**
Dr Charles Bitamazire Businge;
cbusingae@gmail.com

## ABSTRACT

**Background** Although subclinical hypothyroidism in pregnancy is one of the established risk factors for pre-eclampsia, the link between iodine deficiency, the main cause of hypothyroidism, and pre-eclampsia remains uncertain. We conducted a systematic review to determine the iodine nutrition status of pregnant women with and without pre-eclampsia and the risk of pre-eclampsia due to iodine deficiency.

**Methods** MEDLINE, EMBASE, Google Scholar, Scopus and Africa-Wide Information were searched up to 30th June 2020. Random-effect model meta-analysis was used to pool mean difference in urinary iodine concentration (UIC) between pre-eclamptic and normotensive controls and pool ORs and incidence rates of pre-eclampsia among women with UIC <150 µg/L.

**Results** Five eligible studies were included in the meta-analysis. There was a significant difference in the pooled mean UIC of 254 pre-eclamptic women and 210 normotensive controls enrolled in three eligible case–control studies (mean UIC 164.4 µg/L (95% CI 45.1 to 283.6, p<0.01, $I^2$ >50)). The overall proportions of pre-eclampsia among women with UIC <150 µg/L and UIC >150 µg/L in two cross-sectional studies were 203/214 and 67/247, respectively, with a pooled OR of 0.01 (95% CI 0 to 4.23, p=0.14, $I^2$ >50) for pre-eclampsia among women with UIC >150 µg/L. The overall incidence of pre-eclampsia among women with UIC <150 µg/L and UIC >150 µg/L in two cohort studies was 6/1411 and 3/2478, respectively, with a pooled risk ratio of 2.85 (95% CI 0.42 to 20.05, p=0.09, $I^2$ <25).

**Conclusion** Although pre-eclamptic women seem to have lower UIC than normotensive pregnant women, the available data are insufficient to provide a conclusive answer on association of iodine deficiency with pre-eclampsia risk.

**PROSPERO registration number** CRD42018099427.

## INTRODUCTION

Subclinical hypothyroidism is a risk factor for pre-eclampsia, which is a prominent cause of maternal and perinatal morbidity and mortality.[1–3] Iodine deficiency, which is exacerbated by pregnancy-related physiological changes, is a leading cause of hypothyroidism.[4 5] Hence, among women within the reproductive age bracket, insufficient

### Strengths and limitations of this study

► The current study is among the first systematic reviews that have ascertained the relationship between insufficient iodine nutrition status and pre-eclampsia.
► This review has however been limited by the small number of eligible studies coupled with small sample sizes.
► The varied study designs coupled with a considerable degree of heterogeneity precluded the pooling of all the results.

nutrition status prior to the onset of pregnancy, which potentially worsens during the course of pregnancy, could increase the risk of pre-eclampsia like is the case for fetal neurological complications, particularly in endemic iodine deficiency settings.[6 7]

Over two billion people live in areas with iodine insufficiency.[8] Iodine deficiency is on the rise in areas originally thought to be iodine sufficient, despite concerted worldwide efforts to promote iodine fortification. This is partly attributed to high concentration of perchlorate and thiocyanate in water sources and the diet, which impairs the uptake of iodine by the thyroid gland, particularly among individuals with thyroid stimulating hormone (TSH)-related thyroid stimulation secondary to iodine deficiency, and to ineffective implementation and monitoring of dairy-based and bread-based iodine supplementation strategies.[9–12]

In this systematic review and meta-analysis, we sought to establish if there is a difference in the urinary iodine concentration (UIC) of pregnant women with and without pre-eclampsia and whether pregnant women with insufficient iodine nutrition status are at increased risk of pre-eclampsia.

The study is reported according to the Preferred Reporting Items for Systematic reviews and Meta-Analysis guidelines[13] and

was based on a protocol that was registered with the International Prospective Register of Systematic Reviews.

## METHODS
### Eligibility criteria
#### Inclusion criteria

The selection of studies for inclusion was guided by the Population, Intervention/Exposure, Comparison and Outcome protocol. The target population was pregnant women, and the exposure was insufficient iodine nutrition status before pregnancy for cohort studies and insufficient iodine nutrition status during pregnancy for case–control studies. The iodine nutrition status was defined according to the WHO/International Council for Control of Iodine Deficiency Disorders classification of iodine intake using median UIC.[14 15] For pregnant women, a UIC <150, 150–249, 250–499 and >500 μg/L are considered an estimate of insufficient, adequate, more than adequate and excessive iodine nutritional status, respectively.[15] The comparators were study participants with sufficient iodine nutrition status (UIC ≥150 μg/L) during pregnancy.[14 15] The outcomes were the prevalence and incidence rates of pre-eclampsia among women with and without adequate iodine nutrition status from which the ORs for case–control and risk ratios for cohort studies were determined.

Pre-eclampsia was defined as new-onset hypertension after 20 weeks of amenorrhoea characterised by elevated systolic blood pressure (SBP >140 mm Hg) and/or diastolic blood pressure (DBP ≥90 mm Hg), based on two measurements 4 hours apart, or SBP >160 mm Hg and/or DBP >110 mm Hg from a single measurement. Elevated blood pressure (BP) had to be accompanied by at least one of the following: proteinuria in 24 hour urine ≥300 mg or protein/creatinine ratio ≥0.3 mg/mg or urine protein measured by dipstick ≥2+, thrombocytopenia (platelet count less than $150 \times 10^9$L), kidney insufficiency (serum creatinine levels above 90 μmol/L), decreased liver function (AST and ALT twice higher than the upper limit of the reference interval), compromised lung function or pulmonary oedema, visual or other symptoms and signs of impaired cerebral function.[16] There may be considerable heterogeneity if pre-eclampsia has been variably defined in different studies that are eligible for inclusion in the current systematic review.

#### Exclusion criteria

Studies were excluded if they lacked means, medians, ORs, incidence and prevalence rate data to compute them even after repeated unsuccessful attempts to contact the authors via email for relevant information. Letters to editors, reviews, commentaries, editorials and any publication without primary data were also excluded.

### Patient and public involvement

There was no involvement of the public or patients.

### Search strategy and selection criteria

We searched PubMed, Scopus, Web of Science, Academic Search Premier, Africa-Wide Information, CINAHL, Cochrane Library, Google Scholar and Health Source: Nursing/Academic Edition databases for all published studies on iodine deficiency and pre-eclampsia up to 30th June 2020. This search was conducted using a predefined comprehensive and sensitive search strategy (table 1) combining relevant terms and synonyms, which are variably used to denote abnormally high BP in pregnancy and insufficient iodine intake or iodine deficiency as detailed in the published protocol for this review.[17]

### Study selection and data extraction

Two authors (CBB and AU) independently screened the titles and abstracts of identified studies. Citations and abstracts were initially screened and duplicate citations excluded. Titles and abstracts were then screened following inclusion criteria described in the protocol,[17] after which

**Table 1** Search strategy for MEDLINE[15]

| Population: **pregnant** women with pre-eclampsia | | |
|---|---|---|
| #1 | MeSH terms | Pregnant Women [Mesh] OR Pregnancy [Mesh] OR Pregnancy Trimesters [Mesh] |
| #2 | Free text | Pregnancy OR Pregnant women OR expectant mothers |
| #3 | #1 OR #2 | |
| #4 | MeSH terms | Pre-Eclampsia [Mesh] OR Eclampsia [Mesh] OR Hypertension [Mesh] |
| #5 | Free text | Preeclampsia OR Pre-eclampsia OR Eclampsia OR Hypertension OR Hypertensive OR High blood pressure |
| #6 | #4 OR #5 | |
| Exposure: iodine deficiency | | |
| #7 | MeSH terms | Iodine [Mesh] |
| #8 | Free text | Iodine |
| #9 | #7 OR #8 | |
| #10 | #3 AND #6 AND #9 | |

MeSH, Medical Subject Headings.

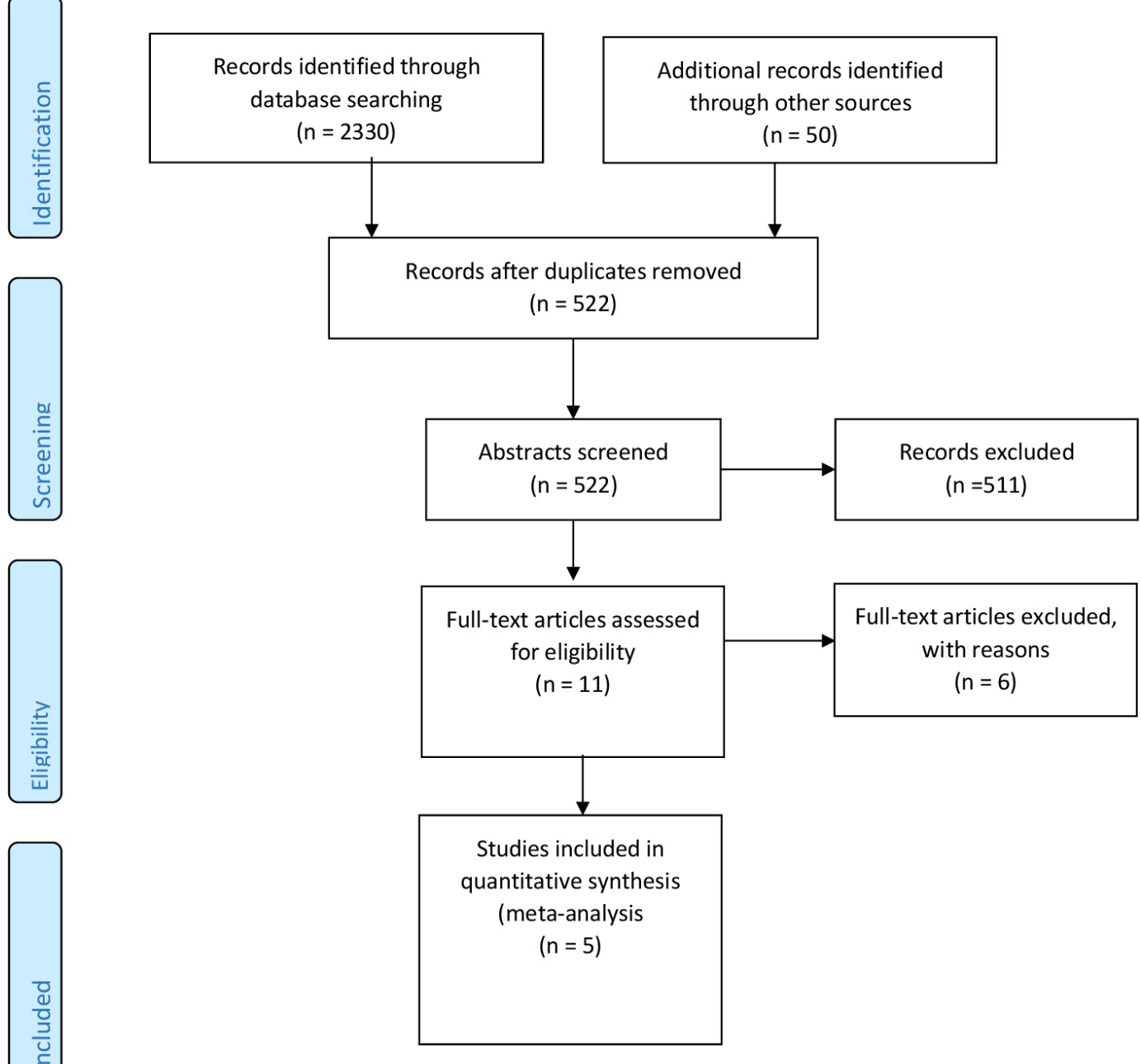

**Figure 1** Study selection flow chart.

the full texts of potentially eligible articles were obtained. These full texts were screened using a standardised and pretested form to include eligible studies. Disagreements were resolved by consensus. For each study, one reviewer (CBB) extracted the data, and a second reviewer (AU) checked the accuracy. For the five studies included here, there were no disagreements between the two reviewers. Figure 1 shows the flow chart for the selection process. The following data were extracted from the eligible studies: study characteristics (authors, years, design and study regions), study population (age and sample size), iodine nutrition status of the various study groups and the methods of outcome measurement.

### Quality assessment

Two reviewers (CBB and AU) independently scored the risk of bias and the quality of included studies (table 2A,B) using the Newcastle-Ottawa Scale.[18] Inter-rater agreement on screening, data abstraction and methodological quality (selection, comparability of groups and ascertainment of exposure/outcome) was assessed using Cohen's κ coefficient.[19] The kappa value for inter-rater agreement for quality assessment was 0.694 (p<0.001). Discrepancies were resolved by consensus.

**Table 2** (A) Risk of bias assessment (reviewer: CBB). (B) Risk of bias assessment (reviewer: AU)

| Study | Study type | Selection | Comparability | Outcome/exposure | AHRQ Scale (good/fair/poor) |
|---|---|---|---|---|---|
| (A) | | | | | |
| [23] | Case–control | *** | ** | *** | Good |
| Cuellar-Rufino et al[24] | Case–control | **** | ** | *** | Good |
| Businge et al[25] | Case–control | **** | ** | ** | Good |
| Yang et al[26] | Cohort | **** | ** | *** | Good |
| [27] | Cohort | **** | ** | *** | Good |
| (B) | | | | | |
| Gulaboglu et al[23] | Case–control | *** | ** | *** | Good |
| Cuellar-Rufino et al[24] | Case–control | *** | ** | ** | Good |
| Businge et al[25] | Case–control | **** | ** | *** | Good |
| Yang et al[26] | Cohort | **** | ** | *** | Good |
| [27] | Cohort | **** | ** | *** | Good |

AHQR, Agency for Healthcare Research and Quality.

## Data synthesis, analysis and assessment of heterogeneity

The analysis was performed with the Review Manager (RevMan) Software, V.5.4 (the Nordic Cochrane Centre, the Cochrane Collaboration) and the 'meta' and 'metafor' packages of the statistical software R (V.4.0.2, the R Foundation for Statistical Computing, Vienna, Austria). For the outcomes of interest (means, prevalence and incidence rates), random-effect model meta-analyses were used to pool estimates across studies with similar design.[20] The degree of heterogeneity across studies was assessed using the Cochrane Q Statistic and Inconsistency Index ($I^2$) (statistics and values ranked as indicating low, $I^2 < 25\%$; moderate, 25%–50%; and high heterogeneity, $I^2 > 50\%$).[21] The Egger funnel plot was used to check for publication bias.[22]

## RESULTS

### The review process

The process for selecting the relevant studies is summarised in figure 1. In total, 2380 records were identified via database searches. After removing duplicates, we scanned the titles and abstracts of 522 articles, of which 11 full texts were further reviewed. Of these, five articles met criteria for inclusion in the current systematic review.

### Characteristics of included studies

All the five included studies were categorised as having a low risk of bias. Their characteristics are summarised in table 3. Three were institutional-based case–control studies one from the countries Turkey, Mexico and the Democratic Republic of Congo, while two were prospective cohort studies that were from two different provinces (Henan and Liaoning) in China.[23–28]

## Meta-analysis

### Mean difference in UIC of pre-eclamptic and normotensive women

Three studies reported the mean UIC of pre-eclamptic and normotensive pregnant women.[23–25] Overall, there was a significant and positive mean difference in UIC and standardised mean UIC of normotensive pregnant women and pre-eclamptic women, with substantial heterogeneity across studies (figure 2).

### The risk of pre-eclampsia among women with UIC <150 µg/L

Two case–control studies had data with proportions of pre-eclamptic and normotensive participants with UIC above or below <150 µg/L.[22 23] The odds of pre-eclampsia among women with UIC <150 µg/L were above unity for individual studies, but the pooled OR of 86.73 (0.32 - 23 509.12) was not significant with substantial heterogeneity across studies ($I^2=73\%$) (figure 3A,B).

The incidence of pre-eclampsia in the two cohort studies was 2/2320 and 7/1576, respectively.[26 27] There was no difference in the incidence of pre-eclampsia among participants with or without low UIC (<150 µg/L) as shown in figure 4.

### Publication bias

Visual inspection of funnel plot symmetry suggested potential publication bias for the studies included in the meta-analysis of UIC difference of pre-eclamptic and normotensive counterparts as well as the odds of pre-eclampsia among women with UIC <150 µg/L (figure 5).

After adjustment of the effect size for potential publication bias using the trim-and-fill correction, two potentially missing studies (figure 6) were imputed in funnel plot (mean UIC differences of −389.60 (−413.02; −366.17) and −512.50 (−556.23; −468.78), respectively. With potential inclusion of the missing studies, the pooled mean UIC was estimated to be −278.0000 (−438.3025; −117.6975),

**Table 3** Characteristics of included studies

| Study | Country | Study design | Study period | Cases (n) | Controls (n) | Comparator | Diagnostic criteria |
|---|---|---|---|---|---|---|---|
| [23] | Turkey | Case–control | Not stated | Severe pre-eclampsia (40) | Normotensives (18) | Mean UIC | Sandell-Kolthoff reaction |
| Cuellar-Rufino et al[24] | Mexico | Case–control | Jan–April 2015 | Pre-eclampsia (20) | Normotensives (37) | ► Mean UIC<br>► UIC <150 µg/L | Fast colorimetric method |
| Businge et al[25] | Democratic Republic of Congo | Case–control | Jan 2007 to December 2008 | Pre-eclampsia (68) and severe pre-eclampsia/eclampsia (182) | Normotensives (150) | ► Mean UIC<br>► UIC <150 µg/L | Sandell-Kolthoff reaction |
| Yang et al[26] | Henan Province, China | Cohort | July to September 2015 | ► Incident pre-eclampsia 1/718 for women with UIC <150 µg/L<br>► Incident gestational HT 17/718 for women with UIC <150 µg/L | ► Incident pre-eclampsia 1/1602 for women with UIC >150 µg/L<br>► Incident gestational HT 25/1602 for women with UIC >150 µg/L | ► UIC <150 µg/L | Fast colorimetric method |
| [27] | Liaoning Province, China | Cohort | 2012–2014 | ► Incident pre-eclampsia 5/693 for women with UIC <150 µg/L<br>► Incident gestational HT 18/693 for women with UIC <150 µg/L | ► Incident pre-eclampsia 2/876 for women with UIC >150 µg/L<br>► Incident gestational HT 25/876 for women with UIC >150 µg/L | ► UIC <150 µg/L | Sandell-Kolthoff reaction |

HT, hypertension ; UIC, urinary iodine concentration.

which is significantly different from the pooled estimate of the three included studies (p<0.001).

The funnel plot for the cohort studies included in the assessment of the incidence of pre-eclampsia among women with UIC <150 µg/L was not suggestive of potential publication bias (figure 7).

## DISCUSSION

The current review has shown that pre-eclamptic women have significantly lower mean UIC than their normotensive counterparts. This trend was observed in all the three included studies despite being from three different continents: Africa, Europe and South

**Figure 2** Forest plot showing the mean difference in urinary iodine concentration of normotensive and pre-eclamptic mothers.

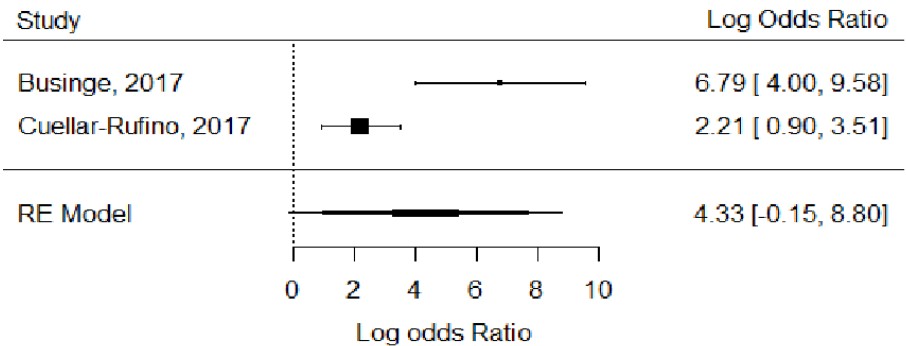

| Study | Experimental Events | Total | Control Events | Total | Weight | Odds Ratio MH, Random, 95% CI |
|---|---|---|---|---|---|---|
| Businge, 2017 | 187 | 187 | 63 | 213 | 36.4% | 4450.67 [8.97; 2207532.11] |
| Cuellar-Rufino, 2017 | 16 | 27 | 4 | 29 | 63.6% | 9.09 [2.46; 33.53] |
| Total (95% CI) | | 214 | | 242 | 100.0% | 86.73 [0.32; 23509.12] |

Heterogeneity: $Tau^2 = 12.4066$; $Chi^2 = 3.66$, df = 1 (P = 0.06); $I^2 = 73\%$

**a**

| Study | | Log Odds Ratio |
|---|---|---|
| Businge, 2017 | | 6.79 [ 4.00, 9.58] |
| Cuellar-Rufino, 2017 | | 2.21 [ 0.90, 3.51] |
| RE Model | | 4.33 [-0.15, 8.80] |

Log odds Ratio

**b**

**Figure 3** (A) Forest plot showing the odds of pre-eclampsia among women with urinary iodine concentration (UIC) <150 µg/L. (B) Forest plot showing the log odds of pre-eclampsia among women with UIC >150 µg/L (p=0.068, $Tau^2$=9.8, $I^2$=88.94%).

America.[23–25] This association between low UIC and pre-eclampsia may reflect inadequate iodine intake predating pregnancy persisting until the third trimester that may increase the risk of pre-eclampsia among susceptible women. A recent Norwegian Study reported that among women with mild to moderate deficiency, long-term preconception iodine supplementation was associated with reduced incidence of pre-eclampsia.[7]

Although there was a trend towards a positive association between low UIC (<150 µg/L) in the third trimester and pre-eclampsia for the included case–control and cohort studies, the pooled OR and risk ratio showed a non-significant association. The small number of eligible studies that also had substantially high heterogeneity may partially account for this result. Hence, the available data are insufficient to provide a definitive answer on the risk of pre-eclampsia associated with low UIC in the third trimester.

Iodine deficiency is thought to predispose to incident pre-eclampsia through two mechanisms. The first one is the reduction of the antioxidant capacity of the placenta, which is one of the organs where the sodium

| Study | Experimental Events | Total | Control Events | Total | Weight | Risk Ratio MH, Random, 95% CI |
|---|---|---|---|---|---|---|
| Xiao 2017 | 5 | 693 | 2 | 876 | 74.1% | 3.16 [0.61; 16.24] |
| Yang 2018 | 1 | 718 | 1 | 1602 | 25.9% | 2.23 [0.14; 35.62] |
| Total (95% CI) | | 1411 | | 2478 | 100.0% | 2.89 [0.42; 20.05] |

Heterogeneity: $Tau^2 = 0.0013$; $Chi^2 = 0.04$, df = 1 (P = 0.83); $I^2 = 0\%$

**Figure 4** Forest plot showing the risk of pre-eclampsia among women with urinary iodine concentration <150 µg/L (designated as experimental group). Pooled risk ratio=2.89 (0.42 to 20.05), p=0.09.

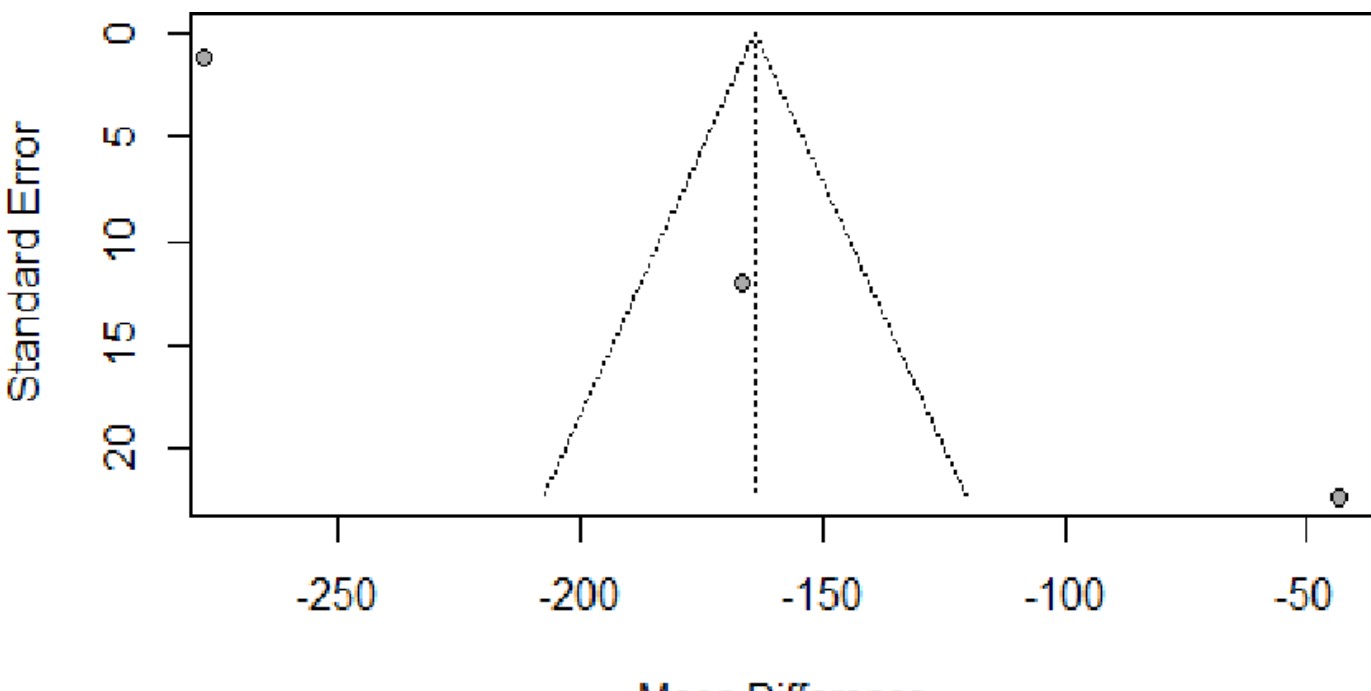

**Figure 5** Funnel plot for the studies selected for the analysis of the mean difference in urinary iodine concentration of pre-eclamptic women and normotensive counterparts.

iodine symporter maintains a high concentration of iodine, which, among other roles, is thought to reduce oxidative stress and lipid peroxide formation, which are elevated in patients with pre-eclampsia.[28 29] The second mechanism is persistent iodine deficiency predisposing to elevated TSH. TSH operating via its endothelial receptors has been shown to diminish endothelial nitric oxide and prostacyclin production as well as upregulate endothelin production, which lead to endothelial dysfunction and systemic vasoconstriction.[30–32] Since baseline prepregnancy as well as serial pregnancy UIC analyses were not carried out in the two cohort studies, it remains uncertain whether the iodine nutritional status at enrolment truly reflected the iodine nutritional status before pregnancy and for the remaining duration of the pregnancy following enrolment. Iodine nutritional status is likely to change with dietary habits and the progressive physiological changes of pregnancy. This could lead to misclassification of study participants and dilute the association between iodine deficiency and pre-eclampsia.[33] The estimation of maternal intrathyroid iodine concentration, even though more technical, has been proposed as a more objective measure of prepregnancy iodine nutrition status than spot UIC.[34] Concurrent measurement of spot UIC and serum thyroglobulin may help identify individuals with long-term

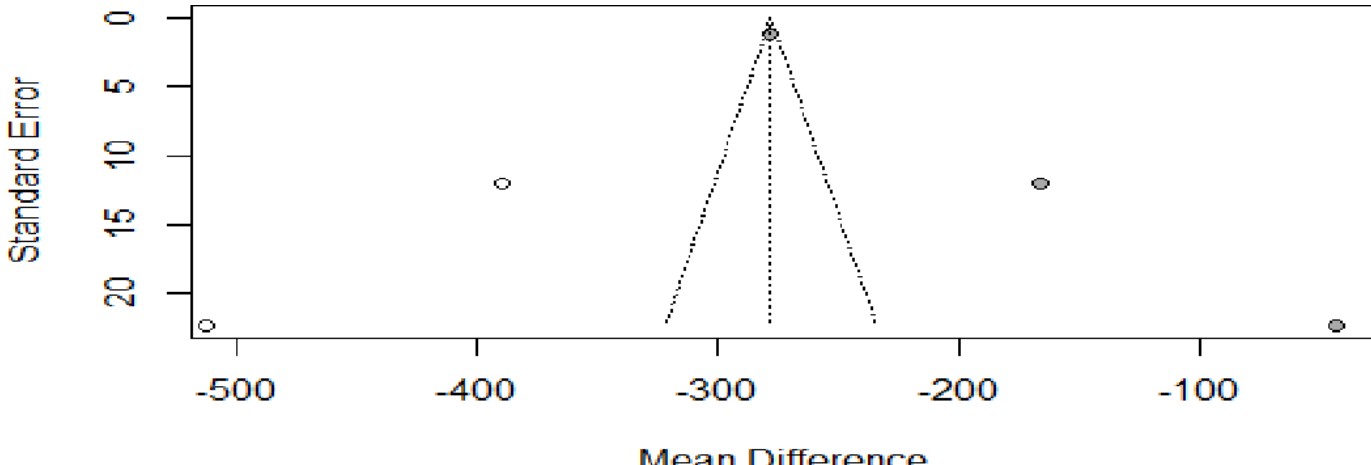

**Figure 6** Funnel plots for publication bias in the studies selected for the analysis of the mean difference urinary iodine concentration of pre-eclamptic women and normotensive counterparts. The two imputed studies are represented by empty circles.

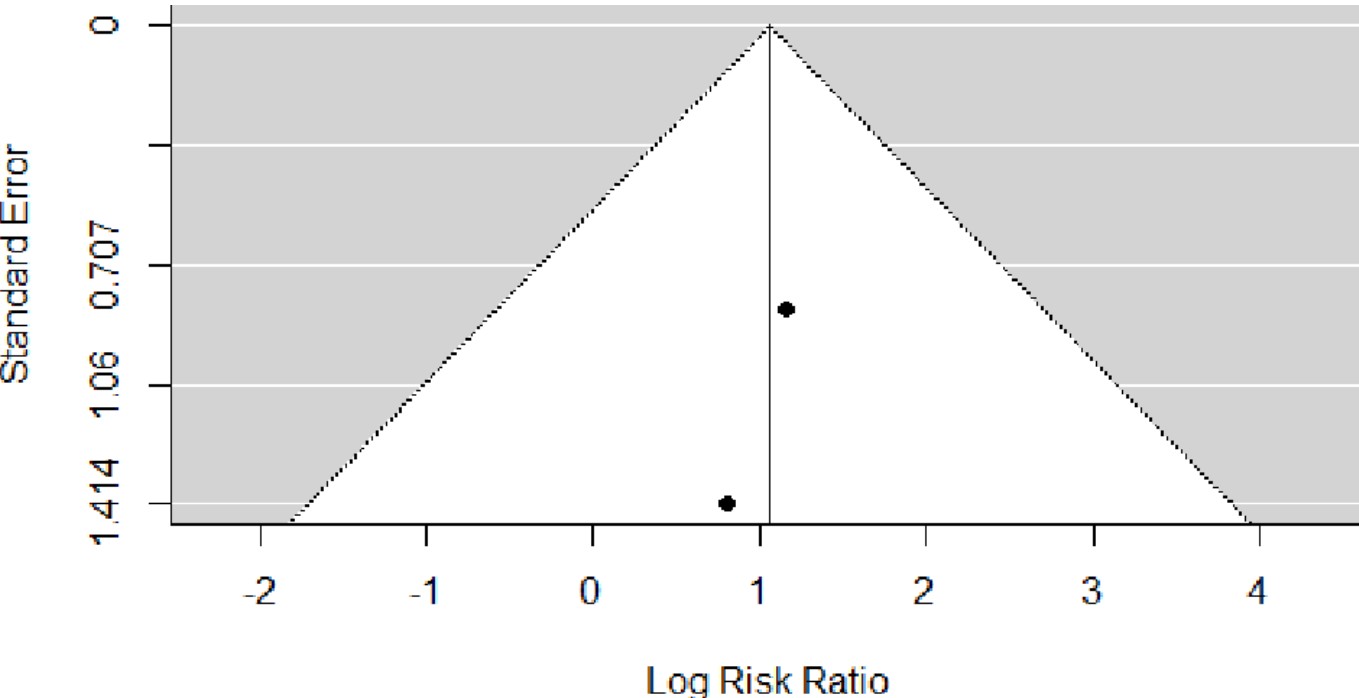

**Figure 7** Funnel plot for the studies selected for the analysis of the risk of pre-eclampsia among women with urinary iodine concentration <150 μg/L.

exposure to iodine deficiency in studies where it is not possible to measure serial UIC and intrathyroid iodine concentration.[35 36]

## Limitations

This review was limited by the small number of eligible studies with small sample sizes and substantial degree of heterogeneity. The varied research designs of the eligible studies precluded the pooling of all the test results.

## CONCLUSION

Although the UIC of women who present with pre-eclampsia seems to be lower than that of women who remain normotensive until delivery, the available data are insufficient to reliably draw a conclusion on the association of iodine deficiency with the risk of pre-eclampsia. More well-designed and adequately powered studies that also include the estimation of prepregnancy iodine nutrition status are needed to address this question.

**Contributors** CBB and APK conceived and designed the study. CBB carried out the literature search. CBB and AU screened the identified studies and extracted data from eligible studies. CBB was responsible for data analysis and writing the first manuscript. APK, AU and BLM took part in critical revision of the first manuscript. All the authors read and approved the final version of the manuscript.

**Funding** This study is part of a research project supported by the Discovery Foundation Rural Fellowship with grant number 038372. APK is employed by the South African Medical Research Council.

**Competing interests** None declared.

**Patient and public involvement** Patients and/or the public were not involved in the design, or conduct, or reporting, or dissemination plans of this research.

**Patient consent for publication** Not required.

**Provenance and peer review** Not commissioned; externally peer reviewed.

**Data availability statement** All data relevant to the study are included in the article or uploaded as supplementary information. All data relevant to the study are included in the article.

**ORCID iD**
Charles Bitamazire Businge http://orcid.org/0000-0002-8393-1198

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
