## [Reviewer comments · BMJ Open]

ARTICLE DETAILS

TITLE (PROVISIONAL)	Insufficient iodine nutrition status and the risk of preeclampsia: a systemic review and meta-analysis
AUTHORS	Businge, Charles; Usenbo, Anthony; Longo-Mbenza, Benjamin; Kengne, AP

VERSION 1 – REVIEW

REVIEWER	Dr Kelly-Ann Eastwood St Michael's Hospital, University Hospitals Bristol and Weston NHS Foundation Trust, United Kingdom.
REVIEW RETURNED	05-Nov-2020

GENERAL COMMENTS	A well written paper posing an interesting research question. The authors have acknowledged the major limitation of the work which is the small numbers of papers eligible for inclusion in the review. In addition, application of the cut offs for iodine deficiency in the cohort studies with few cases of pre-eclampsia and high degree of heterogeneity means a clinically useful conclusion cannot be reached. I would suggest updating the reference for diagnosis of pre-eclampsia: . Tranquilli AL, Dekker G, Magee L, Roberts J, Sibai BM, Steyn W, Zeeman GG, Brown MA. The classification, diagnosis and management of the hypertensive disorders of pregnancy: a revised statement from the ISSHP. Pregnancy Hypertens. 2014;4:97–104. doi: 10.1016/j.preghy.2014.02.001.
--

REVIEWER	Anne Lise Brantsæter Norwegian Institute of Public Health, Oslo, Norway
REVIEW RETURNED	09-Nov-2020

GENERAL COMMENTS	Comments to the authors In this manuscript, the authors report the results of a systematic review to determine the iodine status of pregnant women with and without preeclampsia and the risk of preeclampsia due to iodine deficiency. They did this in order to elucidate the potential link between iodine deficiency and preeclampsia. There were few studies available and the conclusion is that although preeclamptic women have lower urinary iodine concentration than normotensive pregnant women, there is insufficient data to provide a conclusive answer on the association of iodine deficiency with preeclampsia risk. Preeclampsia is a serious complication affecting 2%–8% of all pregnancies worldwide. The aetiology is largely unknown, but given
--

the link between hypothyroidism and preeclampsia, a link with iodine nutrition is to be expected. A major problem is, however, that iodine status is notoriously difficult to assess. A single spot UIC provides a very poor measure of iodine status at the individual level due to fluctuations in intake of dietary iodine and fluid. Therefore, case-control studies are better suited than cohort studies to answer the research question in this review. In cohort studies, using UIC as the exposure requires a large sample size to be powered to detect differences. Another concern is the low degree of iodine deficiency in some of the studies included in the review, as some are from countries with widespread salt iodization and more than adequate iodine nutrition.

Here are some specific issues for the authors to address:

Corrections:

- Page 4, line 31, the number 3 should be spelled out and placed prior to "eligible"
- Page 4 Lines 31 and 33 says "cross-sectional" studies, but in the main text these studies are referred to as case-control studies and denoting them cross-sectional is incorrect. Cross-sectional is also used in the results (page 7, line 170) and this must be corrected.
- Page 4, line 82, correct the unit for UIC from $\mu\text{g } \mu\text{g/L}^{-1}$ to $\mu\text{g/L}$ as used elsewhere in the manuscript
- Page 5, line 84, correct "sufficient iodine status ($>150 \mu\text{g/L}$) before the onset of pregnancy for case-control studies" to ($\geq 150 \mu\text{g/L}$) i.e. include those with $\text{UIC}=150$ as sufficient
- Page 5, line 85, correct "and $>100 \mu\text{g/L}$ before the onset of pregnancy for cohort studies" to $\geq 100 \mu\text{g/L}$ (as above, include $=100$)

Major issues

1. Page 4, lines 56-57. The sentence "As a consequence, insufficient iodine intake in pregnancy could be a risk factor of preeclampsia, particularly in endemic iodine deficiency settings" needs to be corrected because thyroid function in pregnancy depend on the longer-term iodine intake, and not iodine intake in pregnancy. There is no evidence of any beneficial effect of initiating iodine supplement in pregnancy. Women need to enter pregnancy with a sufficient thyroidal iodine store to meet the increased demand during pregnancy 1 2.

It is a widespread misunderstanding that iodine intake in pregnancy can counteract poor iodine intake prior to pregnancy, but as with folic acid, it takes time to build up a sufficient thyroidal iodine store.

2. According to the methods (page 5, line 85), the two cohort studies included in the review compare preeclampsia prevalence in women with $\text{UIC}<100$ and those with $\text{UIC} \geq 100 \mu\text{g/L}$. However, in Table 3 the authors write $\text{UIC}>150\mu\text{g/L}$ vs $\text{UIC} >150 \mu\text{g/L}$. Likewise, in the Legend to Figure 4 (page 10, line 252 and below Figure 4), the authors write "risk of preeclampsia among women with $<150 \mu\text{g/L}$ ". This is very confusing and raises concern as to what the author have actually done. This may also pertain to the Figure 7

	funnel plot. 3. For pregnant women, UIC is categorised using 150 µg/L as the cut-off. WHO recommends UIC for evaluation of iodine intake at the group level. According to WHO, pregnant women with UIC <150 have insufficient iodine intake, denoted mild-to-moderate iodine deficiency. Although initiating iodine supplementation in pregnancy will increase UIC, there is limited scientific evidence for a beneficial impact on health outcomes in mothers or children for this higher cut-off in pregnancy. An important lesson from the era of endemic iodine deficiency was that trials to prevent endemic cretinism only succeeded if the iodine was provided prior to conception (Pharoah et al. 1971, reprinted in IJE 2012)³. Therefore, the cut-off of 100 µg/L for group median is actually a better cut-off when examining pregnancy outcomes as well. I would strongly encourage the authors to elaborate the importance of iodine sufficiency in women of fertile age in the introduction, given the importance of correcting iodine deficiency prior to pregnancy. 4. In spite of being a poor measure of the long-term iodine status at the individual level, it is the only measure available that reflects iodine nutrition at the group level in countries that have multiple iodine sources and where salt iodization has been implemented. In the introduction the authors refer to a rise in iodine deficiency in countries thought to be iodine sufficient for decades. This particularly pertain to countries that have relied on milk and fish as sources of iodine and have no salt iodization. One example is the UK and some Scandinavian countries, which had large areas of endemic goitre until the 1930 ties when mineral salt added to cows' fodder resulted in cow's milk being a major iodine source⁴. In a large pregnancy cohort in Norway, iodine intake from food has been shown to be a reliable measure of iodine intake and a recent study examined the associations between iodine intake and pregnancy outcomes. In this large cohort, iodine intake below the RDI for non-pregnant women was associated with significantly increased risk of preeclampsia ⁵. UIC measured in a subsample of nearly 3000 cohort participants confirm inadequate iodine intake (median UIC 69 µg/L) and using iodine intake as the exposure this study have reported adverse associations between maternal iodine intake and time to pregnancy, fetal growth and neurocognitive development in offspring⁶⁻⁸. When analyses were restricted to the subsample with UIC, the association only reached significance only for fetal growth⁵, illustrating the limitation of UIC. I encourage the authors to elaborate the importance of a sufficient sample size and of more reliable assessment of iodine status than a single UIC in future studies. 5. I fully agree with the authors' statement in the discussion that UIC in third trimester is a poor marker of iodine status and that repeated UICs and serum thyroglobulin may help identifying individuals with long term iodine deficiency. However, as mentioned above, the time of crucial importance for identifying iodine deficiency should be prior to pregnancy, not after conception. I encourage the authors to bring this into the discussion. References 1. Dineva M, Fishpool H, Rayman MP, et al. Systematic review and meta-analysis of the effects of iodine supplementation on thyroid function and child neurodevelopment in mildly-to-moderately
--	---

	iodine-deficient pregnant women. Am J Clin Nutr 2020;112(2):389-412. doi: 10.1093/ajcn/nqaa071. 2. Nazeri P, Shariat M, Azizi F. Effects of iodine supplementation during pregnancy on pregnant women and their offspring: A systematic review and meta-analysis of trials over the past three decades. Eur J Endocrinol 2020 doi: 10.1530/eje-20-0927 [published Online First: 2020/10/29] 3. Pharoah P, Buttfeld IH, Hetzel BS. Neurological damage to the fetus resulting from severe iodine deficiency during pregnancy. Int J Epidemiol 2012;41(3):589-92. doi: 10.1093/ije/dys070. 4. Phillips DI. Iodine, milk, and the elimination of endemic goitre in Britain: the story of an accidental public health triumph. J EpidemiolCommunity Health 1997;51(4):391-93. 5. Abel MH, Caspersen IH, Sengpiel V, et al. Insufficient maternal iodine intake is associated with subfecundity, reduced foetal growth, and adverse pregnancy outcomes in the Norwegian Mother, Father and Child Cohort Study. BMC Med 2020;18(1):211. doi: 10.1186/s12916-020-01676-w 6. Abel MH, Brandlistuen RE, Caspersen IH, et al. Language delay and poorer school performance in children of mothers with inadequate iodine intake in pregnancy: results from follow-up at 8 years in the Norwegian Mother and Child Cohort Study. Eur J Nutr 2019;58(8):3047-58. doi: 10.1007/s00394-018-1850-7. 7. Abel MH, Caspersen IH, Meltzer HM, et al. Suboptimal Maternal Iodine Intake Is Associated with Impaired Child Neurodevelopment at 3 Years of Age in the Norwegian Mother and Child Cohort Study. J Nutr 2017;147(7):1314-24. doi: 10.3945/jn.117.250456
--	---

VERSION 1 – AUTHOR RESPONSE

Reviewer 1

I would suggest updating the reference for diagnosis of pre-eclampsia:

Tranquilli AL, Dekker G, Magee L, Roberts J, Sibai BM, Steyn W, Zeeman GG, Brown MA. The classification, diagnosis and management of the hypertensive disorders of pregnancy: a revised statement from the ISSHP. *Pregnancy Hypertens*. 2014;4:97–104. doi: 10.1016/j.preghy.2014.02.001.

Response:

The reference for diagnosis of pre-eclampsia (currently number 16 instead of 14) has been updated as advised and appropriate changes made in the text.

Reviewer 2

Here are some specific issues for the authors to address:

Corrections:

- Page 4, line 31, the number 3 should be spelled out and placed prior to “eligible”

Response:

This has been corrected and now reads as:

....in three eligible case-control studies.....

- Page 4 Lines 31 and 33 says “cross-sectional” studies, but in the main text these studies are referred to as case-control studies and denoting them cross-sectional is incorrect. Cross-sectional is also used in the results (page 7, line 170) and this must be corrected.

Response

Thank so much for this observation. This has been corrected in all affected sentences where cross-sectional has been replaced with case-control studies.

- Page 4, line 82, correct the unit for UIC from $\mu\text{g L}^{-1}$ to $\mu\text{g/L}$ as used elsewhere in the manuscript

Response

$\mu\text{g L}^{-1}$ was changed to $\mu\text{g/L}$

- Page 5, line 84, correct “sufficient iodine status ($>150 \mu\text{g/L}$) before the onset of pregnancy for case-control studies” to ($\geq 150 \mu\text{g/L}$) i.e. include those with $\text{UIC}=150$ as sufficient

$>150 \mu\text{g/L}$ has been corrected to $\geq 150 \mu\text{g/L}$

- Page 5, line 85, correct “and $>100 \mu\text{g/L}$ before the onset of pregnancy for cohort studies” to $\geq 100 \mu\text{g/L}$ (as above, include $=100$)

Since all enrolment was made in pregnancy even for cohort studies, the statement “and $>100 \mu\text{g/L}$ before the onset of pregnancy for cohort studies” was been deleted.

Major issues

1. Page 4, lines 56-57. The sentence “As a consequence, insufficient iodine intake in pregnancy could be a risk factor of preeclampsia, particularly in endemic iodine deficiency settings” needs to be corrected because thyroid function in pregnancy depend on the longer-term iodine intake, and not iodine intake in pregnancy. There is no evidence of any beneficial effect of initiating iodine supplement in pregnancy. Women need to enter pregnancy with a sufficient thyroidal iodine store to meet the increased demand during pregnancy 1 2.

It is a widespread misunderstanding that iodine intake in pregnancy can counteract poor iodine intake prior to pregnancy, but as with folic acid, it takes time to build up a sufficient thyroidal iodine store.

Response:

Lines 56-57 have been edited. They now read as:

Hence, among women within the reproductive age bracket, insufficient nutrition status prior to the onset of pregnancy, which potentially worsens during the course of pregnancy, could increase the risk of preeclampsia like is the case for foetal neurological complications, particularly in endemic iodine deficiency settings.

2. According to the methods (page 5, line 85), the two cohort studies included in the review compare preeclampsia prevalence in women with $\text{UIC}<100$ and those with $\text{UIC} \geq 100 \mu\text{g/L}$. However, in Table 3 the authors write $\text{UIC}>150\mu\text{g/L}$ vs $\text{UIC} >150 \mu\text{g/L}$. Likewise, in the Legend to Figure 4 (page 10, line 252 and below Figure 4), the authors write “risk of preeclampsia among women with $<150 \mu\text{g/L}$ ”. This is very confusing and raises concern as to what the author have actually done. This may also pertain to the Figure 7 funnel plot.

Response:

Since all participants in the cohort studies were enrolled during pregnancy the statement “and $\text{UIC} >100 \mu\text{g/L}$ before the onset of pregnancy for cohort studies” was deleted.

3. For pregnant women, UIC is categorised using $150 \mu\text{g/L}$ as the cut-off. WHO recommends UIC for evaluation of iodine intake at the group level. According to WHO, pregnant women with $\text{UIC} <150$ have insufficient iodine intake, denoted mild-to-moderate iodine deficiency. Although initiating iodine supplementation in pregnancy will increase UIC, there is limited scientific evidence for a beneficial impact on health outcomes in mothers or children for this higher cut-off in pregnancy. An important lesson from the era of endemic iodine deficiency was that trials to prevent endemic cretinism only

succeeded if the iodine was provided prior to conception (Pharoah et al. 1971, reprinted in IJE 2012)³. Therefore, the cut-off of 100 µg/L for group median is actually a better cut-off when examining pregnancy outcomes as well.

Response:

Thanks so much for this useful information. Most of the included studies already used cut off 150 µg/L

I would strongly encourage the authors to elaborate the importance of iodine sufficiency in women of fertile age in the introduction, given the importance of correcting iodine deficiency prior to pregnancy.

Response

In the introduction (lines 55 – 57), we have stated that:

Hence, among women within the reproductive age bracket, insufficient nutrition status prior to the onset of pregnancy, which potentially worsens during the course of pregnancy, could increase the risk of preeclampsia like is the case for foetal neurological complications, particularly in endemic iodine deficiency settings.

4. In spite of being a poor measure of the long-term iodine status at the individual level, it is the only measure available that reflects iodine nutrition at the group level in countries that have multiple iodine sources and where salt iodization has been implemented. In the introduction the authors refer to a rise in iodine deficiency in countries thought to be iodine sufficient for decades. This particularly pertain to countries that have relied on milk and fish as sources of iodine and have no salt iodization. One example is the UK and some Scandinavian countries, which had large areas of endemic goitre until the 1930 ties when mineral salt added to cows' fodder resulted in cow's milk being a major iodine source⁴. In a large pregnancy cohort in Norway, iodine intake from food has been shown to be a reliable measure of iodine intake and a recent study examined the associations between iodine intake and pregnancy outcomes. In this large cohort, iodine intake below the RDI for non-pregnant women was associated with significantly increased risk of preeclampsia ⁵. UIC measured in a subsample of nearly 3000 cohort participants confirm inadequate iodine intake (median UIC 69 µg/L) and using iodine intake as the exposure this study have reported adverse associations between maternal iodine intake and time to pregnancy, fetal growth and neurocognitive development in offspring⁶⁻⁸. When analyses were restricted to the subsample with UIC, the association only reached significance only for fetal growth⁵, illustrating the limitation of UIC. I encourage the authors to elaborate the importance of a sufficient sample size and of more reliable assessment of iodine status than a single UIC in future studies.

Response

Thank you so much for the information on the importance of dairy products as the major source of iodine in some European Countries. Thank you too for the recent article that also delineates the relationship between low iodine intake, preeclampsia and other adverse pregnancy outcomes. We acknowledged the importance of sufficient sample size in the study limitations (lines 229 – 230) where we stated that:

“This review was limited by the small number of eligible studies with small sample sizes and substantial degree of heterogeneity”,

as well as in lines 236 – 237 in the conclusion where we have stated that:

“More well-designed and adequately powered studies that also include the estimation of pre-pregnancy iodine nutrition status are needed to address this question”.

We agree with the need for a more reliable assessment of iodine nutrition status than a single UIC in

future studies. We have stated in lines 222 – 226 that:

“The estimation of maternal intra-thyroid iodine concentration, even though more technical, has been proposed as a more objective measure of pre-pregnancy iodine nutrition status than spot UIC. xx Concurrent measurement of spot UIC and serum thyroglobulin may help identify individuals with long-term exposure to iodine deficiency in studies where it is not possible to measure serial UIC and intra-thyroid iodine concentration”.

5. I fully agree with the authors' statement in the discussion that UIC in third trimester is a poor marker of iodine status and that repeated UICs and serum thyroglobulin may help identifying individuals with long term iodine deficiency. However, as mentioned above, the time of crucial importance for identifying iodine deficiency should be prior to pregnancy, not after conception. I encourage the authors to bring this into the discussion.

Response

Thank you for this observation.

In the discussion (lines 200 – 205), we have stated that:

This association between low UIC and preeclampsia may reflect inadequate iodine intake predating pregnancy persisting till the third trimester that may increase the risk of preeclampsia among susceptible women. A recent Norwegian study reported that among women with mild to moderate deficiency, long-term preconception iodine supplementation was associated with reduced incidence of preeclampsia.

In addition, in the conclusion (lines 235 – 237) we have stated that:

More well-designed and adequately powered studies that also include the estimation of pre-pregnancy iodine nutrition status are needed to address this question.

VERSION 2 – REVIEW

REVIEWER	Anne Lise Brantsæter Norwegian Institute of Public Health
REVIEW RETURNED	16-Jan-2021
GENERAL COMMENTS	The authors have addressed my comments to my full satisfaction and I have no further comments. The revised text in this review now highlights in a much better way that iodine status needs to be improved prior to pregnancy, which is an important message parallell to the message for folic acid. Thank you!